# Safe Care and Medication Intake Provided by Caregivers at Home: Reality Care Study Protocol

**DOI:** 10.3390/healthcare11152190

**Published:** 2023-08-03

**Authors:** José Joaquín Mira, Pura Ballester, Eva Gil-Hernández, Luisanna Sambrano Valeriano, Esther Álvarez Gómez, Clara Olier Garate, Álvaro Márquez Ruiz, María Acedo Torrecilla, Almudena Arroyo Rodríguez, Ezequiel Hidalgo Galache, Paloma Navas Gutiérrez, Virtudes Pérez-Jover, Susana Lorenzo Martínez, Irene Carrillo Murcia, César Fernández Peris, Alicia Sánchez-García, María Asunción Vicente Ripoll, Ángel Cobos Vargas, Pastora Pérez-Pérez, Mercedes Guilabert Mora

**Affiliations:** 1Departamento Psicología de la Salud, Universidad Miguel Hernández, 03202 Elche, Spain; v.perez@umh.es (V.P.-J.); icarrillo@umh.es (I.C.M.); c.fernandez@umh.es (C.F.P.); suni@umh.es (M.A.V.R.); mguilabert@umh.es (M.G.M.); 2Health District Alicante-Sant Joan, 03013 Alicante, Spain; 3Atenea Research Group, Foundation for the Promotion of Health and Biomedical Research, 03550 Sant Joan d’Alacant, Spain; maria.ballester04@umh.es (P.B.); eva.gilh@umh.es (E.G.-H.); alicia.sanchezg@umh.es (A.S.-G.); 4Departamento Farmacología, Universidad Católica de San Antonio de Murcia, 30107 Murcia, Spain; 5El Escorial Hospital, 28200 Madrid, Spain; luisanna.sambrano@salud.madrid.org (L.S.V.); estherag@hotmail.com (E.Á.G.); paloma.navas@salud.madrid.org (P.N.G.); 6Hospital Universitario Fundación Alcorcón, 28922 Alcorcón, Spain; clara.olier@salud.madrid.org (C.O.G.); slorenzom@salud.madrid.org (S.L.M.); 7Hospital y Residencia San Juan Grande, 11408 Jerez de la Frontera, Spain; alvaro.marquez@sjd.es; 8Hospital Universitario Dr. Peset, 46017 Valencia, Spain; acedo_mar@gva.es; 9Centro Universitario de Enfermería San Juan de Dios, Universidad de Sevilla, 41930 Bormujos, Spain; almudena.arroyo@sjd.edu.es; 10Ramón y Cajal Universitary Hospital, 28034 Madrid, Spain; ezequiel.hidalgo@salud.madrid.org; 11Hospital Universitario Clínico San Cecilio, 18016 Granada, Spain; angel.cobos.sspa@juntadeandalucia.es; 12Unidad Territorial II. Provincia San Juan de Dios de España, 41005 Sevilla, Spain; pastora.perez@sjd.es

**Keywords:** patient safety, virtual reality, 360-degree videos, caregivers, dwelling

## Abstract

Justification: Providing care to patients with several conditions and simultaneously taking several medications at home is inexorably growing in developed countries. This trend increases the chances of home caregivers experiencing diverse errors related with medication or care. Objective: To determine the effectiveness of four different educational solutions compared to the natural intervention (absence of intervention) to provide a safer care at home by caregivers. Method: Prospective, parallel, and mixed research study with two phases. Candidates: Home-based caregivers caring a person with multiple comorbid conditions or polymedication who falls into one of the three profiles of patients defined for the study (oncology, cardiovascular, or pluripathological patients). First phase: Experts first answered an online survey, and then joined together to discuss the design and plan the content of educational solutions directed to caregivers including the identification of medication and home care errors, their causes, consequences, and risk factors. Second phase: The true experiment was performed using an inter- and intrasubject single-factor experimental design (five groups: four experimental groups against the natural intervention (control), with pre- and post-intervention and follow-up measures) with a simple random assignment, to determine the most effective educational solution (n = 350 participants). The participants will be trained on the educational solutions through 360 V, VR, web-based information, or psychoeducation. A group of professionals called the “Gold Standard” will be used to set a performance threshold for the caring or medication activities. The study will be carried out in primary care centers, hospitals, and caregivers’ associations in the Valencian Community, Andalusia, Madrid, and Murcia. Expected results: We expect to identify critical elements of risk management at home for caregivers and to find the most effective and optimal educational solution to reduce errors at home, increasing caregivers’ motivation and self-efficacy whilst the impact of gender bias in this activity is reduced. Trial Registration: Clinical Trial NCT05885334.

## 1. Introduction

### 1.1. Background and Rationale

The increase in the number of people with chronic diseases is a significant burden of care and consumption of health and care resources all over the world [1]. Most are elderly and, in some degree, dependent on a caregiver for their diary activities. Thanks to the work of caregivers caring for people at home, these people can survive while preserving their well-being as much as possible [2]. While this protocol covers a wide range of activities, ranging from physical and emotional support to fostering autonomy to the management of pharmacological treatment and caregiving [3], it is important to acknowledge that the results cannot be considered conclusive at this stage. Instead, it could be suggested that the results may have a significant impact. Throughout this introduction, we will address aspects related to gender bias and other gaps, such as the generational gap and the digital divide, which are especially relevant when considering the elderly population. The gender bias generally prompts that the provision of care for dependent persons is carried out by women, which widens the already existing gender gap to the detriment of women [4]. Treatments and care at home are already more complex, demanding more than the basic tasks of hygiene and feeding; however, caregivers usually receive instructions from healthcare professionals over a limited period of time [5].

To the best of our knowledge, some attempts have been made to alleviate the lack of training in caregiving through caregiver schools, self-help groups, information in brochures, websites, equipment design, or apps with problem-solving tools [6]. All these approaches have in common the caregivers’ placement as a passive entity, as a recipient of information and training on home care through traditional methods. Moreover, some sources of information exist; however, there is a lack of quality control on their contents. Regarding medication, the resources/developments that are currently available include (1) passive information in brochures, websites, or apps to prevent errors; (2) redesign of medication packaging to avoid confusions; (3) development of dispensing devices seeking ease and safety usage by caring for the patient experience; (4) patient education campaigns, empowering patients about their medication; and (5) medication alarms adapted to mobiles or tablets [7,8]. However, as far as we are concerned, it is unknown what the impact of active learning in appropriate treatment administration and reduction of errors by caregivers at home will be and if it would be the solution [9]. Studies have been highlighted the frequency and type of medication errors at home. Less information is available about care errors made by these caregivers [7].

There are several ways of learning certain tasks. One can learn by listening to information about a procedure, following some written instructions, by watching a video, or by being immersed in a virtual environment where actively learning a process is possible and perceived by users as positive [10]. Exploiting the potential of immersive active learning technologies such as virtual reality (VR) or 360-degree video (360 V) would allow the caregiver to be placed at the center of their learning [11]. VR and 360 V have been successfully used for the training of healthcare professionals in adjuvant treatments in clinical situations or to break the isolation of people living in institutions or alone [12]. Nevertheless, 360 V and VR have hardly been used to provide caregivers with the skills and self-confidence to deal with medications, caring tasks, and errors in their roles in homes.

### 1.2. Protocol Definitions

*Medication errors* refer to any avoidable incident that may cause inappropriate use of medication or harm to the patient, while the drug is under the control of the healthcare professional, patient, or caregivers.

*Care errors* describe any avoidable incident during care, grooming, or mobilization that can cause or does cause harm to the patient and that, in our case, occurs in the home and is performed by caregivers.

*Qualified caregivers* are those who have received a specific training or qualification, for 20 h or more, whose content is centered in strategies or knowledge related to caring to a patient. *Informal caregivers* are those who have not acquired any qualification related with the tasks mentioned above.

### 1.3. Objectives

The main purpose of this study is to compare the success of different caregivers’ educative solutions and determine which is the most suitable solution to reduce medication and care errors at home, increasing caregivers’ motivation and self-efficacy. The effects of training through 360 V, VR, web-based information, psychoeducational intervention, and standard procedures will be compared to determine which is the best way to train home caregivers to increase their technical competencies and soft skills, including self-confidence. Previously, this study explored the experience and needs of caregivers (qualified and informal) of dependent and chronically ill patients and then to design and test the effectiveness of several interventions with the goal of at promoting safer medication use in the home.

### 1.4. Trial Design

The study design is a prospective, parallel, and mixed true experiment, with five experimental arms involving three different types of caregivers that attend to cardiovascular, oncological, or multi-pathological (having more than two comorbid chronic conditions, e.g., diabetes and dementia) and polymedicated patients, and consisting of three phases. The project seeks to put people at the center of interventions while respecting their digital rights. A true experimental design will be conducted to determine the effectiveness of four educational solutions to train same home care interventions. Each of them plus a control constitutes an arm, and their ability to reduce errors during the care and medication administration at home by caregivers will be assessed. This study uses a simple random assignment of participants to one of five arms. It will compare inter-group and intra-subject PRE and POST measures. The three groups of caregivers will follow this experiment with an itinerary of training adapted to their recipients’ needs.

### 1.5. Protocol Phases

#### 1.5.1. Phase I: Deciding the Focus of the Experimental Intervention

First, nominal groups will be used to gather information on knowledge, needs, and differences in the responsibility for the safe use of medication and dispensing of care among caregivers. Additionally, an online survey involving members of caregiver associations was disseminated via email. Both approaches allowed the identification of errors, causes, consequences, and risk factors that favor these errors. Thanks to all this information, home situations in which it was particularly important to train caregivers in medication dispensing or providing care were identified. This information fed into the design of treatment and care-providing scenarios for the experimental phase.

#### 1.5.2. Phase II: Defining the Education Paths of the True Experiment

Then, a panel of expert clinicians from seven hospitals voted through an online poll. There, they could inform about the treatment and care-providing scenarios that will be more pertinent in their centers considering the caregivers’ profile and identified gaps in their training. Furthermore, the participating centers filled in that poll using the sample of participants that they could enroll and chose the educational solutions that will be trained in their centers according to their requirements.

With the result of the previous online polls, three different profiles of caregivers will be selected based on the type of patient they are caring for. The rationale behind participants falling into these three categories is that, on the one hand, caregivers of oncology patients need to be trained in specific devices that these patients have, such as ostomies or a Port-a-Cath. They need to learn how to change, clean, maintain, or manipulate them without causing the patients any harm. Patients with a cardiovascular problem need to have a caregiver familiar with blood pressure measurements or sudden changes of weight caused by an edema, to name a few. Finally, a group of diverse conditions have been grouped into the last group, multiple comorbidities, where patients can have ameliorated movements, dementia, or diabetes. Therefore, training in administration of multiple medications and hygiene is needed for caregivers.

The educational solution content consisted in seven different treatment and care scenarios of care applicable to the average recipient in each profile (Table 1).

For example, in the cardiovascular training package, there are four scenarios to train the special care needed if the patient experiences an ictus (e.g., patient mobilization from bed to wheelchair, hygiene in bed, cognitive stimulation, and changing of diapers) and three scenarios related to medication (e.g., injected medication administration, errors in oral medication administration and usage, and constant vital measurements).

#### 1.5.3. Phase III: True Experiment Development

A total of five different educational solutions were proposed, corresponding to the arms of the true experimental study. The performance of the different education solutions will be assessed. The process of this phase of the study is summarized in Figure 1.

As mentioned above, some centers may not participate in some arms, and they will all be specialized in one or two, at maximum, types of caregivers.

## 2. Methods

### 2.1. Study Setting

The study will be carried out simultaneously in seven hospitals, from three different autonomous regions (the Valencian Community, Andalusia, and the Community of Madrid) in Spain.

### 2.2. Participant Eligibility Criteria

The study candidates will be either qualified or informal caregivers who oversee patients in one of three groups: (1) multi-pathological and polymedicated, (2) oncological, or (3) diagnosed cardiovascular disease patients.

The inclusion criteria in this protocol for caregivers are (1) their patient must have either multiple simultaneous diagnoses (e.g., diabetes, hypertension, etc.) and (>4) ongoing medications, or an active oncological illness, or a diagnosed cardiovascular disease (e.g., cerebrovascular ischemic event or heart failure), with a high dependency rate according to a Barthel score equal or lower than 55; (2) the candidate caregiver must supervise this patient at least 6 months per year (3 months in the case of oncology patients); and (3) they must be attending in a hospital located in the Communities of Valencia, Andalusia, or Madrid.

The exclusion criteria are common to all groups of caregivers: (1) filed a patrimonial claim in the last 5 years; (2) have experience in using VR or AR for a similar purpose; (3) over 90 year of age; (4) attending to a ratio of patients higher than 2; (5) suffer from problems of vertigo, tinnitus, motion sickness, epilepsy, seizures or similar symptoms, severe cardiac conditions, or wearing a cardiac pacemaker or hearing aid.

Additionally, during the recruitment, two aspects must be considered: (1) the *qualification of the caregivers*, and a balanced sample of qualified and informal caregivers will be sought, as much as possible, in both the control and four experimental groups: (2) *gender* of the caregivers, forcing at least 30% of the study sample to be male caregivers.

### 2.3. Sample Size

The sample size of 350 participants (70 in each group) was calculated to detect differences on 10 percentage points in the questionnaires administered to caregivers, considering results of self-efficacy measures and error frequency, for a confidence level of 95%, a statistical power of 80%, and dropout rate of 15%.

### 2.4. Recruitment

Caregivers will be informed about the study, what their commitment is, what is needed from them, and the requirements. Those who are interested and meet the inclusion and exclusion criteria will be asked to sign the consent form.

Recruitment will be carried out in the collaborating centers. Several channels will be used; the protocol information will be delivered through complex chronic patient training schools for caregivers and associations or foundations for caregivers, and also using the “snowball” technique. Participants (caregivers) may be recruited during a regular medical consultation where they go to accompany the person they care for, or in one of their own consultations, where they can state that they are home caregivers.

Subsequently, the centers will randomly assign participants to one of the study arms and construct the registration code for each caregiver. Recruitment will be conducted through simple randomization from the diaries of new patients seen in the last 5 months in the collaborating centers that meet the inclusion criteria. The recruitment technique consists of systematic sampling according to the list of people seen in the consultation and using k = 3. The collaborator will systematically select 1 out of every 3 patients cited in the consultation list. If the person refuses to be included in the study, the next person on the list will be invited and then k = 3 will be applied. By simple randomization, participants will be assigned to the groups following them giving informed consent.

### 2.5. Participant Timeline

#### 2.5.1. Start-Up Visit (Pre-Intervention)

First, caregivers will be informed about the study and sign an informed consent form. Then, an anonymization code will be created, and the participants will be randomized to one of the three arms (see Figure 2). At this point, the caregivers will go to the medical center to which they belong, where they will be recorded carrying out their simulated caregiver’s work (except for Arm 1—*Control Group*). The recording will be made using a device consisting of a headband with a small camera to allow the participant to record comfortably. At the beginning of the recording, the caregiver’s identification code must be shown. In the same session, at the end of the recording, the centers will explain to the participants that they must fill in a questionnaire with their data, the Pre-intervention Caregiver’s Notebook, which can be accessed from a QR code/link that will be given to them at that moment. Subsequently, the videos of the start-up sessions of each participant will be recorded.

#### 2.5.2. Intervention

In each of the three caregivers’ profiles, key patient care situations will be selected (at least 4 from Table 1) and the corresponding educational solutions. At this point, the corresponding intervention will be carried out for each participant. Depending on the arm to which they have been assigned, the procedure and duration of the intervention sessions will be different. The information and scheme for each intervention is explained in the following sections. 

To be able to determine which educational solutions are optimal, the study has the following arms:*Arm 1*—*Control Group*: Natural intervention (provision of information and answering of questions). This arm will evaluate the natural learning during the action.*Arm 2—Experimental Group 2*: Intervention with written materials and demonstration videos specifically designed for this study. This arm will evaluate the effect of a traditional, more passive form of learning, with little interaction with the content. As an example, caregivers of an oncology patient will have access to a website where they will have written and video materials for all the seven scenarios described in Table 1. The same will apply for the other two caregiver profiles.*Arm 3—Experimental Group 3*: The psychoeducational intervention specifically designed for this study. With this specific intervention, the participants will learn by sharing information with the session facilitator and other participants, which will facilitate a more active form of learning, with a higher degree of interaction than the previous arm. Here, caregivers will attend two sessions together with 10 other caregivers who care for the same profile of patients; one session will be clinically oriented and supervised by a clinician, the other will be conducted by a member of the team expert in psychoeducational intervention. There, caregivers will share their experiences and learn from each other. Both sessions will last 1.5 h each.*Arm 4-Experimental Group 4*: The 360-degree immersion intervention designed specifically for this study. Participants enrolled in this arm will experience a form of learning that is assumed to be more participatory and immersive than conventional videos, as they can interact with the content by changing the camera angle. For example, caregivers of a patient with multiple comorbidities will have access to 7 immersive videos described in Table 1; the caregiver will visualize those videos for 45–60 min. The same procedure will apply in the other two profiles of caregivers.*Arm 5-Experimental group 5*: The virtual reality intervention specifically designed for this study. Using VR to learn procedures will encourage caregivers to learn in a fully immersive and participatory environment where they will have to interact with the virtual content, enhancing their learning experience. Here, caregivers of a patient with cardiovascular problems will have access to 7 scenes in virtual reality according to the content described in Table 1; the caregiver will visualize those videos for 45–60 min. The same procedure will apply in the other two profiles of caregivers.

#### 2.5.3. Post-Intervention Visit

Once each arm’s intervention has been conducted, the participants will be called to participate in a new recording of the care work in the medical center. As in the pre-intervention visit, at the beginning of the recording, the caregiver’s identification code must be shown. At the end, a new link and QR with the Post-intervention Caregiver’s Notebook will be shared with them so that they can fill it in online through the link. This procedure will be applied for all arms except for the control arm (Arm 1), where no recording will be carried out and no post-intervention caregiver’s notebook will be completed.

#### 2.5.4. Follow-Up Visit (+6 Months or 3 Months for Caregivers of Oncology Patients)

After 3 or 6 months (follow-up phase), the participants will be recorded making two attempts at the care they were trained in. As in the previous recordings, at the beginning of the recording, the caregiver’s identification code must be shown. At the end, they will be given a new QR and link to the Caregiver’s Notebook Follow-up.

### 2.6. Measures: Experiment Evaluation

#### 2.6.1. Pre-Intervention Measures

In this first set of questions, there are three blocks of information, the first of which collects the demographic and occupational information of the caregivers. A questionnaire in health alphabetization to assess caregivers’ knowledge [13], together with some crucial aspects regarding medication and care such as caregivers’ perception of causes of a spontaneous adverse event [14] or medication prescriptions will follow [15]. Finally, the caregivers’ emotional status [16], and motivation through the Zarit questionnaire [17], will be measured.

#### 2.6.2. Post-Intervention Measures

Participants from Arms 2–5 will reply to this set of questions. Here, we will again assess the caregivers’ emotional status [16], and motivation through the Zarit questionnaire [17]; in addition, we will evaluate the experience of a chronic care recipient and their caregiver [18]. Furthermore, some questions developed by the research team were created as a Likert scale to assess the caregivers’ satisfaction related to the educational intervention used.

#### 2.6.3. Follow Up-Intervention Measures

Participants from all arms will reply to this set of questions. Here we assessed the permanence of the learning and long-term outcomes related with the educational intervention used. Some measures were reevaluated in order to properly evaluate the ability of the study to reduce care and medication errors including to assess caregivers’ health alphabetization [13], self-awareness of causing a spontaneous adverse event [14], following medication prescriptions [15], emotional status and motivation [16], the Zarit questionnaire [17], and IEXPAC [18]. The total cost of each intervention will be calculated by adding up the cost of personnel, materials, and equipment, to name a few. Additionally, an estimation of the cost of caregivers’ time will be included.

#### 2.6.4. Video Recordings of the Performance in Each Stage

The effectiveness evaluation of each arm will be measured, among other ways, by viewing the video recordings of the two attempts to execute the care situation covered by the education solution. These recordings will be made at three different times: at the baseline visit (pre-intervention), after the intervention of the arm (post-intervention), and 6 months (3 in the case of oncology patients) after the end of the intervention (follow-up). The control Group is an exception, as there is no specific intervention and therefore the pre- and post-intervention recording will be omitted. Thus, only the follow-up intervention after 3–6 months will be recorded.

These video recordings will be assessed by means of an objective checklist (rubric). For each learning situation, the team has created a rubric which lists the typical errors in the care or medication procedure, marking the learning points on which the educational solutions have an impact. These rubrics have been validated by external researchers who received the script of each situation together with the rubric. They assessed the coherence between what has been developed from the script into training educational solutions and what is being assessed in the rubric. Members of the research team will watch and correct, first the videos of the Gold Standard group to set a bar, and then the caregivers’ recordings. This task will be carried out by members of the research team.

Additionally, these same scenarios were performed by a group of professionals (e.g., practitioners, nurses, etc.) who perform these actions daily, to set the standard threshold for the procedure. The study includes a group of 14 healthcare professionals, who will act as the Gold Standard group. This group will be recorded performing all the care situations covered by the education solutions to have a standard measure with which to compare the performance of the caregivers once they have received the training. Candidates for the Gold Standard group must have a university degree in healthcare and been employed for at least five years in a profession where they are taking care of patients of one of the three profiles. In this group, a balanced representation of both genders will be forced.

### 2.7. Data Management

Type of data/research outputs: The data will be observational by nature. Reality Care uses qualitative and quantitative data collected using nominal groups and VR devices. The contributions of those who participate in the study will be recorded (after receiving informed consent from the participants and recipients) in audio and video. Responses to the questionnaires (discrete and continuous variables) will be also registered. Participants will have access to the recording and own data and will be able to request deletion of their interventions at any time. All data will be recorded separately from any personal data. Images that allow the identification of the participants and recipients will be distorted. Primary data will be protected to safeguard the original information obtained and to eliminate the risk of uncontrolled data storage on local computers. The data will not be combined with other existing ones. The data will be processed following GDPR principles (General Data Protection Regulation) on sensitive data management. The ELIXIR Research Data Management Kit (RDMkit) will be used as a guide to assure lawfulness, fairness and transparency, limitation and minimization, accuracy, storage limitation, integrity, and confidentiality. These requisites include the data management life cycle for all data sets that will be generated, collected, or processed during the project, and even after the project is completed. The data will be findable, accessible, interoperable, and reusable according to the FAIR principles. The principal researchers assure the safe storage and management of data and ensure its quality. When considering the sensitive nature and some other privacy issues, special attention will be given regarding subjects’ data including qualitative and quantitative analyses, scale responses, and unstructured information that has been provided, learning outcomes, etc.

### 2.8. Statistical Methods

The time of use of each intervention will be recorded, and correlation analyses will be performed to determine the size and direction of the effect on the outcomes. The differences between the measures before and after the experimental intervention sessions (pre- and post-intervention (time 0) and follow up (+3/+6 months) will be considered as the outcome variables. The distribution of the data will be tested using the Kolmogorov–Smirnov test; the two or three measures (control or other arms, respectively) will be compared with an ANOVA test for normally distributed continuous variables, and the Kruskal–Wallis test if the distribution is not normal. Some categorical variables from questionnaires will be assessed by contingency analyses to compare potential variations before and after the educational solution. Given the number of time-ordered and repeated variables per subject, they will be explored using a linear mixed effects model (LMM), where the age of caregivers will be considered. The data will be stratified considering gender, age, recipient profile, and time of use. *p* values < 0.05 will be considered as statistically significant differences. Statistical analyses will be performed using R and SPSS. If the intervention is successful, the Control Group will be invited to receive the treatment of the experimental group on the same terms. The results of this project will be prepared for dissemination according to the contents of the “Standards for reporting qualitative research” (SRQR) guidelines for qualitative studies, the Consort checklist for randomized trials, and the CHEERS checklist for economic outcome evaluations.

### 2.9. Data Monitoring

The data will be treated according to national and EU regulations. The DMP on privacy/confidentiality and the procedures will be implemented for data collection, storage, protection, retention, and destruction. Furthermore, Reality Care does not collect or process more personal data than is necessary for the performance of the service. We will always obtain the necessary authorizations for collecting and processing the data and—where appropriate—additional free and fully informed consent from the subjects. The application architecture will always be implemented by separating the data layer from application logic and other architectural components to ensure flexibility for data usage and structure. The collection, storage, management, and evaluation of the data will be mainly oriented to evaluate the intervention. All researchers have received standards of the procedure to assure that caregivers are trained homogeneously in all recruitment centers, and among the different experimental arms. Furthermore, the central group (FISABIO) will oversee the monitoring of the online responses from all centers, across the different pre- and post-intervention and follow-up measures, to guarantee the appropriateness of all data collected. The analysis of all video recordings will be carried out by FISABIO with an objective rubric, ensuring a standard evaluation of the intervention results.

### 2.10. Ethics and Dissemination

#### 2.10.1. Research Ethics Approval

The present protocol underwent the revision and approval of an Ethics Committee Board, obtaining approval in December 2021 and January 2023 (CODES 21/063; 22/079 and 22/080). The study has been registered in ClinicalTrial as NCT05885334.

#### 2.10.2. Confidentiality

All the personal data added to the central database will be anonymized. The tools and technologies we will use are de-facto open-source standards that avoid the vendor-lock-in trap. All data will be stored to ensure GDPR compliance and searchability of data/research outputs. After the data are curated and anonymized, selected parts of the data and results can be published via the SD Submit service with a DOI handle. Data curation and anonymization will be performed in data collecting hubs.

#### 2.10.3. Access to Data

The data and results can be shared with other researchers, healthcare professionals, and technicians of caregivers’ associations in a secure desktop environment. The data can be published with open access. Reuse will be guaranteed by providing the data on the OSF repository, Open European Research Platform, and project website; where appropriate, the data or information will be shared upon receiving a reasonable request.

#### 2.10.4. Dissemination Policy

Reality Care is aligned with the open science policy. The findings of the study will be published in scientific journals, and furthermore, the information will be shared with a wide and diverse audience through caregivers’ and health or care professionals’ associations and health policy makers. The materials developed, including the 360° videos, virtual reality scenarios, written information on the website, and psychoeducational issues, will be accessible to other researchers, professionals, and caregivers. By collaborating with healthcare and social welfare authorities, the information will be distributed to numerous caregiver training centers and platforms.

## 3. Discussion

This project responds to the need for efficient caregiver training formats. The demands for home care are increasing, in complexity and type of activity, as a new care economy emerges due to the aging population. Platforms, companies, and family members need support to increase their confidence in providing appropriate care. In healthcare facilities, there is a wide diversity of approaches to training home caregivers, but those based on verbal or written instructions are in the majority. The technologies being compared in this study could reach many people at a low cost. Not all interventions are likely to be suitable for all the care scenarios addressed in this study. The cost analysis framework will enable us to determine which solution is more efficient.

Changes in family structures, higher rates of care dependency, and evolving care needs, coupled with the rising employment rate of women in certain countries, have led to increased demand for care work. In 2015, there were 2.1 billion people in need of care (200 million older people who had reached or exceeded healthy life expectancy). By 2030, the number of care recipients is projected to rise to 2.3 billion, driven by an additional 100 million older people [19]. The number of individuals providing care for others has gradually increased in developed countries [20,21]. This experience is spreading to other countries as policies promoting living at home for as long as possible become more widespread [22]. Home care responsibilities are becoming increasingly complex, and family caregivers require more training. This has led to the emergence of a new care economy. Identifying training methods aimed at providing caregivers and their recipients with greater security is an urgent task [23].

The expected usefulness of the study is based on the principle that personal home care workers are often under-trained, under-resourced, underpaid or not paid at all, and are often used to compensate for the shortage of healthcare workers [24]. Developing teaching solutions with content tailored to certain profiles of caregivers and care recipients, together with the assessment and comparisons of the educational solutions, has been pointed out as a necessity [25] and is the basis for the present work. One of the expected outputs of this project is the detection of the areas where each type of solution can add more value to training (e.g., choosing the right medication vs. properly setting up the materials before a care task or correctly using a Port-a-Cath), as previous studies have already mentioned [26]. Furthermore, in a real context, the recipient could also use some devices to improve self-care, even in the case where the devices will only be used by their caregivers. What is certain is that they will experience the consequences.

The present study will be evaluating the outcomes of the intervention based on the developed questionnaires, together with the rubric we developed to assess the performance of caregivers in each arm. With the assessment of their medication knowledge, we may discover some gaps of information regarding medication as in previous publications [13,14,15]. Previous studies on caregivers have already used the Zarit short version questionnaire to assess some symptoms of discomfort of caregivers, such as depression [27]; we will use this scale together with the motivation scale to assess potential factors that may be worsening their skill development [16,17].

The present protocol is in line with other studies that have raised the importance of increasing the knowledge and skills of home caregivers. Some authors have evaluated the effect of a psychoeducational program on the caregiving competence, including problem-solving coping abilities or mood affections experienced on the job. With a sample of 128 caregivers, after the intervention, there was a significantly greater improvement in terms of caregiving competence, problem-solving coping abilities, and social support satisfaction, to name a few [28]. Other authors have used a high-fidelity simulation-based tracheostomy education program in a programmable mannequin, and for three simulation scenarios: desaturation, mucus plugging, and dislodgement. Thirty-nine caregivers practiced, developing increased confidence through training and finding the training scenarios realistic and helpful [29]. Another randomized clinical trial that enrolled 48 family caregivers proved that an intervention based on a nurse following up over a month and information based on life quality improved the care after the intervention [30]. Despite all the similarities found between the present protocol and previously available educational solutions, this study adds the comparison among different methods of teaching the same content in terms of caregivers’ self-efficacy, problem resolution capacity, or reduction in number of errors in medication or care.

This study will not make it possible to identify all the consequences (especially in the long term) of possible caregiving errors. The strengths of some tools such as VR or 360-degree videos have not been extensively exploited in the field of home care. The results in this study should not be generalized to other contexts (e.g., institutionalized patients), although they will provide guidance on actions that can be implemented to improve caregiver training to increase patient safety. In addition, the studies require the cooperation of caregivers, who should be matched with the appropriate educational solution content. The cost of adverse events that occur and those that are prevented due to the intervention is challenging to calculate, and it cannot be reliably included. Finally, another limitation that this study may face is the generational gap and the digital divide, where some caregivers may have a more positive attitude to the interventions related with technology, and therefore present better, but biased, results. This gap is intended to be reduced by the exclusion criteria of the previous usage of virtual reality. Furthermore, since we have the caregivers’ age, it will be considered a variable when performing the statistical analyses.

The lessons learned from this study may be applicable to other types of care recipients, as there are other pathologies that can present a regimen of care like those proposed in this project (e.g., patients with a cognitive deterioration can have care needs that resemble those proposed for multiple comorbidities). Furthermore, the training materials can serve as a support tool in residential facilities to train and upgrade the knowledge of their staff, especially during vacation periods when is typical to have changes in the workforce.

## 4. Conclusions

The former research will allow to distinguish among different educational approaches and point out which is the best for caregivers’ learning. The findings could play a crucial role in pinpointing the most optimal educational strategies to effectively train caregivers in conducting home care with an enhanced focus on safety. There exists a potential variance wherein specific types of caregiving might benefit from particular educational approaches over others. This underscores the significance of incorporating a wide array of patient profiles and day-to-day scenarios within the scope of the study. These contributions have the potential to indirectly foster the growth of the emerging care economy, granting caregivers the capacity to assume more intricate caregiving roles. This, in turn, could amplify their belief in their own capabilities and offer heightened reassurance to both families and patients. Additionally, given that caregiving responsibilities disproportionately affect women, who often don′t receive the recognition they deserve, the implementation of regulated training serves to diminish their uncertainty while also addressing the gender gap.

## Figures and Tables

**Figure 1 healthcare-11-02190-f001:**
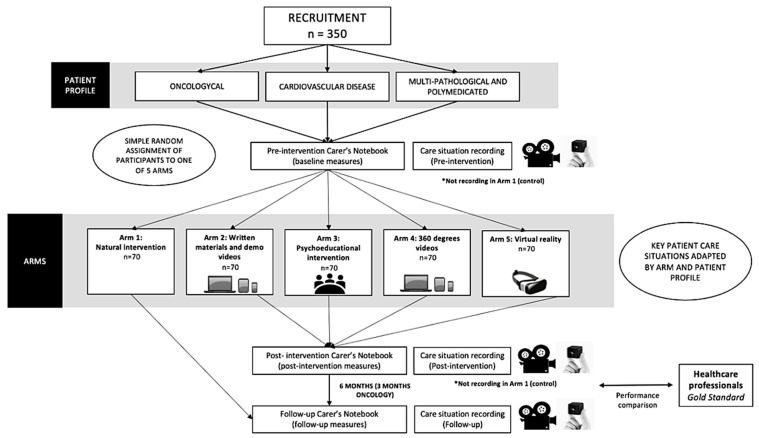
Flowchart of Phase III development of the true experiment. [*] Not video recording.

**Figure 2 healthcare-11-02190-f002:**
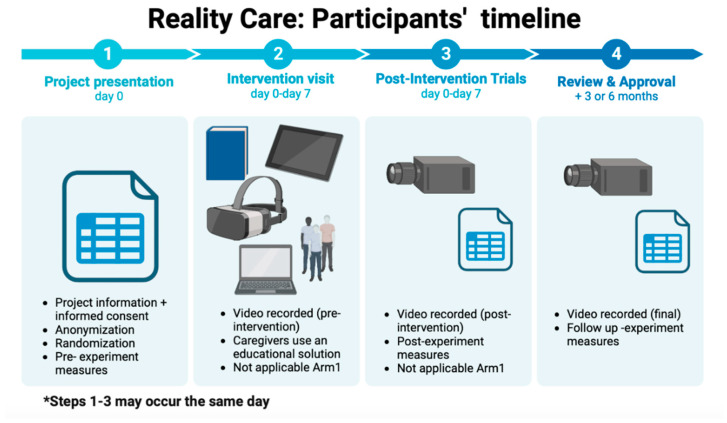
Participants’ timeline in Reality Care protocol.

**Table 1 healthcare-11-02190-t001:** Table of caregiver’s profiles and content.

Recipient Profile	Training Treatment Care Scenarios
Oncology	Port-a-Cath care Administration of injected medications Home quiz Hand sanitizing Diaper changes for bedridden patient Hygiene for people with limited mobility Palliative care at home
Cardiovascular (ictus or heart failure)	Administration of injected and oral medications Home quiz Hand sanitizing Diaper changes for bedridden patient Hygiene for people with limited mobility Resuscitation and defibrillator usage Account for pressure values Protocol for heart failure patients (dietary indications and weight control) Mobile stimulation Deglutition problems Paresthesia stimulation
Multi-morbidities and polymedicated	Administration of injected medications Home quiz Diaper changes for bedridden patient Hip rehabilitation procedures Orthosis Deglutition problems Posture exercises for caregivers during patient transfers Care to prevent pressure injuries

## Data Availability

The data from this study will be available in a freely accessible database, where appropriate, the data or information will be shared upon receiving a reasonable request.

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
