# Peer review of "Safe Care and Medication Intake Provided by Caregivers at Home: Reality Care Study Protocol"

_healthcare, 2023, doi:10.3390/healthcare11152190_

Round 1
Reviewer 1 Report
The research paper addresses a pertinent and timely subject: the growing trend of home caregiving for patients with multiple co-morbid conditions and simultaneous medications in developed countries. The study's focus on identifying critical risk management elements and assessing the effectiveness of educational solutions to enhance caregivers' capabilities is commendable.
However, to strengthen the research, I suggest addressing the following areas through further clarification and consideration:
1. It would be beneficial to provide a more detailed discussion of the specific outcome measures used to evaluate the effectiveness of the educational solutions.
2. The paper mentions caregivers of patients falling into three profiles: oncology, cardiovascular, or pluripathological patients. It is crucial to elaborate on the rationale behind selecting these specific patient profiles and how their inclusion will contribute to the study's overall findings and relevance.
3. An in-depth discussion about how potential biases and confounding factors were identified and addressed during data collection and analysis is essential to ensure the rigor and validity of the study's results.
Based on these minor revisions, I recommend accepting the paper as it has the potential to make a valuable contribution to the field of home caregiving and patient safety.
The English language is fine. Minor editing required.
Author Response
First, we would like to thank the reviewers for the time and effort spent in reviewing this manuscript. We sincerely appreciate their recommendations and insightful comments. They have all provided us with the opportunity to improve the quality of our manuscript.
We have addressed them below, and in the manuscript, as indicated. In addition, we would like to thank the reviewers in advance for their time in reviewing this new version, as well as their final consideration of it.
All the new additions have been marked in yellow to facilitate their identification. Thank you.
REVIEWER 1
The research paper addresses a pertinent and timely subject: the growing trend of home caregiving for patients with multiple co-morbid conditions and simultaneous medications in developed countries. The study's focus on identifying critical risk management elements and assessing the effectiveness of educational solutions to enhance caregivers' capabilities is commendable.
However, to strengthen the research, I suggest addressing the following areas through further clarification and consideration:
- It would be beneficial to provide a more detailed discussion of the specific outcome measures used to evaluate the effectiveness of the educational solutions.
Reply: Reviewer 1 is right and further information regarding the outcome measures has been added in the discussion section.
“The present study will be evaluating the outcomes of the intervention based in developed questionnaires, together with the rubric own developed to assess the performance of caregivers in each arm. With the assessment of their medication knowledge, as in previous publications we may discover some gaps of information regarding medication (13-15). Previous studies in caregivers have already used the Zarit short version questionnaire to assess some symptoms of discomfort of caregivers, such as depression (27), we will use this scale together with the motivation scale to assess potential factors that may be worsening their labor development (16-17)”.
- The paper mentions caregivers of patients falling into three profiles: oncology, cardiovascular, or pluripathological patients. It is crucial to elaborate on the rationale behind selecting these specific patient profiles and how their inclusion will contribute to the study's overall findings and relevance.
Reply: Thank you for this suggestion, that information is now added in the main text, please see Method section, protocol phases, phase II.
“The rational behind participants falling into these three categories is that, on one hand, caregivers of oncology patients need to be trained in specific devices that these patients have, such as ostomies or port-a-cath. They need to learn how to change, clean, maintain or manipulate them without causing patients any harm. Patients with a cardiovascular problem need to have a caregiver familiar with blood pressure measurements, or sudden changes of weight caused by an edema, to name a few. Finally, a group of diverse conditions have been grouped in the last group, multi-morbidities, where patients can have ameliorated movements, dementia, or diabetes. Therefore, a training in multiple medication administrations and hygiene is needed for caregivers”.
- An in-depth discussion about how potential biases and confounding factors were identified and addressed during data collection and analysis is essential to ensure the rigor and validity of the study's results.
Reply: We would like to thank reviewer 1 for his/her appreciation, this information is now added in the section 2.13 of methods.

Reviewer 2 Report
This is an article that outlines the protocol of a project to evaluate four interventions of different nature in various populations. It has the potential for publication; however, there are certain aspects that raise doubts for me. I believe they could be taken into account to enhance its quality and improve comprehension for the reader.
· The sentence: "This work covers a range of activities from physical and emotional support for the promotion of autonomy to the management of their pharmacological treatment and providing care (3). If it is a protocol, it cannot be stated yet that the results will be conclusive. It could be indicated that the results could have an impact. Additionally, they would be included at the end of the introduction.
· In the introduction, there is a mention of sex bias. What do they refer to? Is it gender bias?
· Gender bias refers to discrimination or prejudice based on a person's gender, whereas sex bias specifically focuses on biological differences between males and females.
· Apart from the gender gap, what about the generational gap? Or the digital divide? This is particularly significant when dealing with elderly individuals.
· The second paragraph is written in an informal style. Please review it for formality.
· While this protocol covers a wide range of activities, ranging from physical and emotional support to foster autonomy, to the management of pharmacological treatment and caregiving (3), it is important to acknowledge that the results cannot be considered conclusive at this stage. Instead, it could be suggested that the results may have a significant impact. Throughout this introduction, we will address aspects related to gender bias and other gaps, such as the generational gap and the digital divide, which are especially relevant when considering the elderly population.
Methods:
· Please clarify the meaning of multi-pathological. Define it.
· The interventions are not sufficiently detailed. It is necessary for them to be so detailed that they can be replicated in other studies. The interventions described in the methods section should be presented with enough detail to allow other researchers to replicate them accurately. Providing comprehensive information is crucial for validation and further expansion of the findings in future studies.
Author Response
First, we would like to thank the reviewers for the time and effort spent in reviewing this manuscript. We sincerely appreciate their recommendations and insightful comments. They have all provided us with the opportunity to improve the quality of our manuscript.
We have addressed them below, and in the manuscript, as indicated. In addition, we would like to thank the reviewers in advance for their time in reviewing this new version, as well as their final consideration of it.
All the new additions have been marked in yellow to facilitate their identification. Thank you.
REVIEWER 2
This is an article that outlines the protocol of a project to evaluate four interventions of different nature in various populations. It has the potential for publication; however, there are certain aspects that raise doubts for me. I believe they could be taken into account to enhance its quality and improve comprehension for the reader.
- The sentence: "This work covers a range of activities from physical and emotional support for the promotion of autonomy to the management of their pharmacological treatment and providing care (3). If it is a protocol, it cannot be stated yet that the results will be conclusive. It could be indicated that the results could have an impact. Additionally, they would be included at the end of the introduction.
Reply: Thank you for this insightful comment, this information has now been added in the introduction.
- In the introduction, there is a mention of sex bias. What do they refer to? Is it gender bias? Gender bias refers to discrimination or prejudice based on a person’s gender, whereas sex bias specifically focuses on biological differences between males and females.
Reply: Reviewer is right, we have changed sex bias, for gender bias, since is the second the one we are referring to.
“The gender bias generally prompts that the provision of care for…”
- Apart from the gender gap, what about the generational gap? Or the digital divide? This is particularly significant when dealing with elderly individuals.
Reply: This has been included in the limitations of the study. Thanks for highlighting this point.
- The second paragraph is written in an informal style. Please review it for formality.
Reply: The second paragraph of the manuscript has been re-written with a formal style. Thank you for the suggestion.
- While this protocol covers a wide range of activities, ranging from physical and emotional support to foster autonomy, to the management of pharmacological treatment and caregiving (3), itis important to acknowledge that the results cannot be considered conclusive at this stage. Instead, it could be suggested that the results may have a significant impact. Throughout this introduction, we will address aspects related to gender bias and other gaps, such as the generational gap and the digital divide, which are especially relevant when considering the elderly population.
Reply: We have added that wording in the first paragraph of the introduction and it has significantly improved the clarity of the content. Thank you.
Methods:
- Please clarify the meaning of multi-pathological. Define it.
Reply: That definition has been added in the trial design section.
multi-pathological (having more than 2 comorbid chronic conditions e.g., diabetes and dementia)
- The interventions are not sufficiently detailed. It is necessary for them to be so detailed that they can be replicated in other studies. The interventions described in the methods section should be presented with enough detail to allow other researchers to replicate them accurately. Providing comprehensive information is crucial for validation and further expansion of the findings in future studies.
Reply: Further information has been given in the methods section. Please, see page 7.

Round 2
Reviewer 2 Report
Thank you very much for the changes made to the manuscript. this has improved substantially